# Risk Factors and Clinical Outcomes after Antegrade Intramedullary Nailing in Proximal Humeral Fractures: Insights and Implications for Patient Satisfaction

**DOI:** 10.3390/jpm13081224

**Published:** 2023-08-01

**Authors:** Maximilian Willauschus, Sebastian Grimme, Kim Loose, Johannes Rüther, Michael Millrose, Roland Biber, Markus Gesslein, Hermann Josef Bail

**Affiliations:** 1Department of Orthopedics and Traumatology, General Hospital Nuremberg, Paracelsus Medical University, Breslauer Straße 201, 90471 Nuremberg, Germanyjohannes.ruether@klinikum-nuernberg.de (J.R.); hermann-josef.bail@klinikum-nuernberg.de (H.J.B.);; 2Department of Trauma Surgery and Sports Medicine, Garmisch-Partenkirchen Medical Centre, 82467 Garmisch-Partenkirchen, Germany; 3Department of Traumatology, Clinic Dr. Erler gGmbH, 90429 Nuremberg, Germany; r.biber@erler-klinik.de

**Keywords:** proximal humeral fractures, intramedullary nail, complications, patient-reported outcome measures, health-related quality of life

## Abstract

Background: Proximal humeral fractures (PHFs) are common injuries that can lead to significant functional impairment. This retrospective cohort study aimed to evaluate the clinical outcomes and complications associated with the use of the Targon PH+ (Fa. Aesculap, Germany) intramedullary nail for the treatment of PHFs. Methods: A subgroup consisting of 70 patients with a mean follow-up of 4.91 years out of 479 patients who underwent treatment with the Targon PH+ intramedullary nail for PHFs at a single center between 2014 and 2021 were included. Patient-reported outcome measures (PROMs) and health-related quality of life (HRQoL) were assessed using validated German versions of the Disabilities of the Arm, Shoulder, and Hand (DASH) questionnaire, American Shoulder and Elbow Surgeons (ASES) score, Oxford Shoulder Score (OSS), and EuroQol 5-Dimension 5-Level (EQ-5D-5L). Radiographic assessment was performed using pre- and postoperative imaging. Results: Among the 70 patients of the subgroup who completed follow-up, 21.4% experienced complications, including major complications in 15.7% of cases, all of which were revised (revision rate of 15.7%). Anatomical reduction was achieved in 48,5% of cases. The mean DASH, ASES, and OSS scores were 25.4 ± 22.0, 76.2 ± 21.1, and 38.8 ± 10.3, respectively. Significant correlations were observed among the PROMs, indicating their convergent validity. Additionally, a significant correlation of all used PROMs and patient well-being (HRQoL) was observed. Severe complications and revisions were associated with significantly lower ASES scores (−11.1%, *p* = 0.013). There was a tendency for PROM scores to slightly decline with increasing fracture complexity, although this trend did not reach statistical significance. Our findings indicate that patients over the age of 65 years tend to exhibit lower scores in PROMs and HRQoL measures. Conclusion: The use of the Targon PH+ intramedullary nail for the treatment of PHFs resulted in satisfactory clinical outcomes and acceptable complication and revision rates. The PROMs and HRQoL measures indicated varying levels of disability and symptoms, with major complications, revision surgery, and age negatively impacting shoulder function after midterm follow-up.

## 1. Introduction

Proximal humerus fractures (PHFs) rank among the most prevalent fractures in the elderly population (>65 years), with a notably higher incidence in females, approximately twice that of males [1].

The higher prevalence of fractures in females can be attributed to the increased likelihood of developing osteoporosis in postmenopausal women. Reduced bone stability in this population leads to an elevated susceptibility to fractures, even with low-energy traumas [2].

Despite the high incidence of these fractures, there remains a significant lack of consensus regarding the appropriate treatment options for various types of PHF.

Against the background of a variety of surgical treatment options for displaced fractures of the surgical neck of the humeral head, intramedullary nailing (IMN) is a widely used technique yielding good to excellent outcomes for patients [3,4,5,6]. Antegrade interlocking nail osteosynthesis is a surgical technique used in the management of proximal humeral fractures. It involves the insertion of an intramedullary nail through performing a deltoid splitting approach and using locking head fixation screws, providing stable fixation and alignment of the fractured bone segments, which facilitates early mobilization and promotes successful fracture healing.

The evolution of antegrade proximal humerus nailing can be categorized into three generations. The first generation’s minimally invasive approach led to fixation failure due to inadequate stabilization. The second generation improved stability but faced issues with proximal screw loosening because of non-angular stable proximal screw designs. The third generation introduced more secure locking mechanisms, offering fixed angular stable constructs. Designs such as Stryker T2, Synthes Proximal Humeral Nail, and the Targon PH+ featured enhancements in screw design and insertion options. These advances hold promise for better patient outcomes and reduced complications in treating displaced proximal humerus fractures [7].

The current concept of straight antegrade interlocking nails poses some biomechanical benefits. They have a lower lever arm than the lateral locking plate, which may counteract varus displacing forces. Furthermore, compared to the bent proximal humeral nail, the straight design offers several advantages: Medialization of the entry point in shoulder surgery is crucial as it helps preserve the supraspinatus tendon footprint [8]. Furthermore, when the nail is medialized within the humeral head, it maintains a position which effectively enhances resistance against varus forces. Additionally, the medial entry point proves to be valuable in preventing accidental entry into the fracture zone, particularly in cases of proximal humeral fractures with a fractured greater tuberosity. Lastly, the use of a straight nail with a proximal tip efficiently anchors the densest subchondral zone, offering increased stability and protection against varus forces [9,10].

A small incision, soft tissue and nerve sparing technique, immediate postoperative joint mobilization, and a comparably short operation time represent the major advantages [11,12,13].

Until today, there have been several studies and meta-analyses investigating outcomes with the help of subjective (Patient-Reported Outcome Measures (PROMs), Health-Related Quality of Life (HRQoL)) and objective (radiographic assessment) parameters. Hereby, there are established risk factors including insufficient anatomical reduction, four-part fractures, varus head–shaft angle (<130°) and the occurrence of complications [14,15]. A common critique concerning many existing publications is a short follow-up time of only six to 24 months, disregarding significant improvements of shoulder function several years after surgery and the possibility of detecting long-term complications [4,16]. This goes along with a potential overestimation of the effect of surgery on PHFs on quality of life due to recent hospitalization and concomitant burdens such as pain and the rehabilitation process. As most patients would agree, the long-term outcome is a far more important measure.

Given adequate responsiveness to clinical change and the accuracy of these subjective measures, PROMs can help to save resources and manpower necessary for physical examinations [17,18].

The main aim of this study was to evaluate the mid-term patient-reported outcome measures and quality of life after antegrade nailing of proximal humeral fractures. The secondary study goal was to assess individual risk factors for differences in outcome measures.

## 2. Materials and Methods

### 2.1. Selection and Evaluation of Eligible Patients and Inclusion Criteria

In this retrospective cohort study, a total of 479 patients who underwent treatment with the Targon PH+ intramedullary nail (150 mm, Aesculap, B. Braun, Tuttlingen, Germany) for proximal humeral fractures at a single center (Klinikum Nuremberg Sued, Department of Orthopedic Surgery and Traumatology) between 2014 and 2021 were reviewed. Subsequently, exclusion criteria were applied, resulting in 260 patients who met the eligibility criteria for inclusion in this investigation. Among this group, a subset of 70 patients were successfully followed up using patient-reported outcome measures (PROMs).

The inclusion criteria for this study encompassed patients with isolated proximal humeral fractures, a minimum adequate radiological follow-up of more than 1 year, and provision of consent to participate. Exclusion criteria comprised patients with dementia, insufficient clinical data or inadequate radiologic imaging, pathological fractures, open fractures, additional shaft fractures, and the use of a nail length exceeding 150 mm. All patients who participated in the study provided written informed consent.

Patient contact information and general details were obtained from the clinic’s database and hospital information system.

For EQ-5D-5L, we used the official German version 1.1 (translation certificate available) [19].

For PROMs, the following validated German versions were used:German DASH: Germann G. et al. (2003) [20].German ASES: Goldhahn J. et al. (2008) [21].German OSS: Huber W. et al. (2004) [22].

Parts of this dataset have been already published in a previous study about radiologic risk factors of proximal humeral nailing [15] and have been used for radiographic analysis.

### 2.2. Surgical Technique and Postoperative Care

In every case, the decision for IMN was taken by a board of experienced orthopedic trauma surgeons.

Prior to surgery, patients were positioned in a beach chair position on a radiolucent table with a standard armrest. For implantation of the Targon PH+ nail, a deltoid splitting approach at the anterolateral margin of the acromion was performed. After a longitudinal transection of the clavipectoral fascia and the subacromial bursa, the head fragments were reduced by direct or indirect manipulation, depending on the fracture pattern. For this procedure, a wire was used as a joystick for securing the fragments.

After longitudinal splitting of the supraspinatus tendon, a guide pin at the apex of the humeral head was positioned in both planes, and the nail was inserted 3 to 4 mm below the cartilage level. Then, a minimum of three to four locking head fixation screws were inserted into the head fragments, depending on the fracture pattern, the degree of instability, as well as bone quality.

To secure displaced fragments to the interlocking screws, a “rope-over-bitt” procedure was carried out in cases of severely displaced tuberosities (>1 cm). A single or duplicate distal interlocking screw was placed through the nail percutaneously. Board-certified trauma surgeons performed all operations.

An intra- and postoperative X-ray examination was obtained from all patients to verify both reduction and correct positioning of the implant.

The affected extremity was immobilized in an arm sling for one day to control postoperative pain. On the second postoperative day, passive mobilization by trained physiotherapists was started. The arm sling was removed as soon as the patient tolerated the pain. Active-assisted shoulder movement below the pain threshold was then initiated as soon as wound healing was deemed satisfactory. After six weeks and radiological consolidation, weight-bearing was allowed.

Every patient received postoperative standardized anterior–posterior and y-view radiographic imaging for correct fracture reduction and implant position after passive mobilization. Radiographic follow-up after 12 months or longer was obtained from all patients.

### 2.3. Radiologic Fracture Assessment

The pre- and postoperative radiographic imaging was assessed using PACS software, which involved the analysis of plain radiographic images or computed tomography (CT) scans. Fractures were classified based on the number of fractured parts and according to the Neer classification system [23]. The analysis also focused on assessing the quality of postoperative anatomical reduction of the fracture and the pre- as well as postoperative head–shaft angle.

The follow-up radiographs were carefully evaluated by two experienced orthopedic trauma surgeons to identify any documented complications. An anatomical reduction was considered present when certain criteria were met: the head showed neither varus angulation (<130°) nor valgus angulation (>140°), the y-view showed no anterior or posterior tilt of the head exceeding 20°, the dislocation of the greater or lesser tuberosity was less than 3 mm, and the neck-shaft dislocation was less than 5 mm.

### 2.4. Evaluation of Complications and Revision

Early postoperative complications and revisions were evaluated during the hospital stay of the patients. These included reviews of postoperative X-rays and medical records. Late complications and revisions were obtained from hospital charts during visits at our outpatient clinic or if the patient was re-admitted to the hospital, and finally, at the point of follow-up. All complications were organized according to the Clavien–Dindo classification and then subdivided into major (Clavien–Dindo grades III and IV) and minor complications (Clavien–Dindo grades I and II) [24]. Every subsequent surgery of the proximal humeral head apart from elective implant removal was counted as a revision. Implant failure was defined as removal of the nail and conversion to a different osteosynthesis or the secondary implantation of a reverse shoulder prosthesis.

### 2.5. PROMs (DASH, OSS, ASES) and HRQoL (EQ-5D-5L, EuroQol VAS)

In total, 70 subjects agreed to participate in the study by answering EQ-5D-5L, DASH, ASES, OSS, and the additional study questionnaire via a telephone interview.

### 2.6. Statistical Analysis

All data were obtained and analyzed retrospectively. Statistical analysis was performed using IBM SPSS Statistics for Windows (version 28, 1.0.0.1406, IBM Corp., Armonk, NY, USA). Parametric data were analyzed using t-test and ANOVA for multiple samples, nonparametric data via Mann–Whitney U and Kruskal–Wallis analysis. Pearson’s P correlation coefficient was calculated for parametric data and Spearman’s Rho correlation was calculated for PROMs (nonparametric data). Nominal data were analyzed using cross tables and tested with chi-square test and Fisher’s exact test.

All reported *p*-values are two-tailed, with an alpha level <0.05 considered statistically significant. Unless otherwise stated, descriptive results are demonstrated as mean ± standard deviation and range.

## 3. Results

### 3.1. Descriptive Statistics

A total of 70 patients, thereof 47 female and 23 males, were enrolled with a mean age of 74.5 ± 11.6 (29–97) years and mean follow-up time of 4.9 ± 1.9 (1.6–7.8) years (Table 1).

Table 2 presents the classification of complications observed in the patient cases. A total of 15 patients with complications were recorded, accounting for 21.4% of the cases. Among these, 11 were classified as major complications (15.7%), while 4 were categorized as minor complications (5.7%). The revision rate was 15.7%, not counting three additional patients opting for elective implant removal. Moreover, we analyzed the incidence of complications in patients over and under 65 years. Surprisingly, patients over the age of 65 accounted for a higher rate of complications and revisions, but these differences did not reach statistical significance. The elevated overall complication rate in the study was primarily driven by a significantly increased rate of minor complications in patients under 65 years.

Table 3 provides a breakdown of the major and minor complications experienced by the patients. Multiple patients experienced more than one complication.

### 3.2. Fracture Morphology and Radiographic Analysis

Upon evaluation of all 70 patients, anatomical reduction was achieved in 64.3% of cases. Postoperatively, the integrity of the medial hinge was achieved in 78.6% of patients compared to 18.8% preoperatively.

Prior to surgery, 11.9% of fractures exhibited a physiologic head–shaft angle, while 41.8% had a valgus angle and 46.3% had a varus angle. After surgery, 48.5% of patients demonstrated a physiologic head angulation (130°–140°), 32.4% had a valgus angle (>140°), and 19.1% had a varus angle (<130°). Furthermore, the mean absolute head–shaft angle was measured as 131.8 ± 27.9 (range: 68.60°–217.20°) upon admission and 136.0 ± 9.9 (range: 109.7°–157.6°) after surgery.

### 3.3. PROMs and HRQoL Measures

Following IMN during the follow-up period, patients achieved a mean DASH score of 25.4 ± 22.1 (range: 0.00–81.00), indicating the level of disability and symptoms related to upper extremity function. The mean ASES score was 76.2 ± 21.1 (range: 26.6–100.0), representing the level of pain and shoulder function. The mean OSS score was 38.81 ± 10.3 (range: 9.0–54.0), reflecting the level of shoulder-related disability.

According to the DASH interpretation, 45.7% of patients achieved excellent results, indicating minimal disability and symptoms. Additionally, 27.1% of patients achieved satisfactory results, while another 27.1% of patients achieved poor results, indicating varying degrees of disability and symptoms (see Table 4).

In order to examine the association between PROMs, a correlation coefficient was computed. The analysis demonstrated a significant and statistically robust correlation among the evaluated PROMs.

Importantly, this correlation remained consistent when comparing each PROM with EQ-5D-5L (see Table 5).

### 3.4. Influence of Complications and Revisions on PROMs and HRQoL

Patients sustaining major complications experienced significantly lower ASES (−11.1%: 87.3 ± 16.2 > 77.7 ± 17.9, *p* = 0.013) results compared to minor and no complication development.

However, scores of DASH (−12.8%), OSS (+2.9%), EQ-5D-5L index (+6.3%) and EuroQol VAS (−7.5%) did show a significant correlation (DASH: *p* = 0.070; OSS: *p* = 0.123; EQ-5D-5L index: *p* = 0.602; EuroQol VAS: *p* = 0.885).

When examining the influence of revision surgery, again, only ASES scores reduced significantly from 78.3 ± 21.2 to 65.1 ± 17.2 (−16.8%) (*p* = 0.031). Hereby, the increase in DASH (+14.5%) and decrease in OSS (−4.9%) was apparent, yet not significant (*p* = 0.283, *p* = 0.274).

As with complications, HRQoL measures seem to have no relation at all with revision surgery requirement, as EQ-5D-5L index and EuroQol VAS paradoxically showed a non-significant increase (+4.1%, *p* = 0.948 and +3.7%, *p* = 0.631).

### 3.5. Influence of Fracture Complexity

When observing major complications, patients with four-part fractures sustain more than twice as many complications as statistically expected (2.08, *p* = 0.054), and 41.7% of severe complications occurred in four-part fractures. Thus, the four-part complication rate is 35.7% compared to two- and three-part fractures, where only 12.5% of patients sustained a severe complication.

Analogously, the risk for revision surgery increased 2.3-fold (233.43%) in the four-part fracture group, which is a significant finding (*p* = 0.036).

Furthermore, PROMs and HRQoL measures show clear trends towards deterioration in scores with advancing fragment count, especially when considering four-part fractures (see Table 6). Mean DASH score increased with the increase in the number of fragments from two to four. ASES, OSS, and EQ-5D-5L follow the same trend but means reduced with fracture count (see Figure 1).

### 3.6. Age Factor

When comparing the two age groups, namely those over 65 years (*n* = 58) and those under (*n* = 12), in terms of PROMs and quality of life, we observed significant differences in all tested parameters. Patients over 65 years displayed notably reduced scores in DASH (28.6 vs. 9.9; *p*-value = 0.008), ASES (73.8 vs. 88.1; *p*-value = 0.006), OSS (37.6 vs. 44.9; *p*-value = 0.018), and EQ-5D-5L (0.76; *p*-value = 0.019).

While these differences appear to be intuitive, it is crucial to acknowledge a potential confounding factor, namely the high incidence of 4-part fractures in the subgroup of >65 years (*n* = 14) when compared to no 4-part fractures in the <65 years subgroup.

## 4. Discussion

This clinical investigation demonstrates favorable clinical and mid-term results, along with a good overall quality of life, following the use of antegrade nailing for proximal humeral fractures. The study’s outcomes align with the existing scientific literature. Several factors were identified as potential risks leading to a decline in shoulder function, reflected in lower scores in PROMs.

The complexity of the fracture showed a noticeable yet statistically non-significant impact on all applied PROMs as the fragment count increased. Major complications and revision surgery also demonstrated unfavorable effects, particularly evident in reduced ASES scores.

A noteworthy correlation was observed between patients’ quality of life and objective measurements obtained from the employed PROMs. However, the aforementioned factors (fracture complexity, major complications, and revisions) did not significantly influence the overall quality of life.

### 4.1. PROMs and HRQoL

Overall, the clinical outcome measured by PROMs after interlocking nailing of proximal humeral fractures can be considered to be good. This also applies for quality of life in our collective. In this, our data stand in line with the current literature.

Wong et al. reported slightly better results considering ASES (means: 84.4 vs. 76.2) in a large metanalysis with 448 included fractures, on average 22.6 months after intramedullary nailing [25]. Notably, the data by Wong had been collected from a 10-year-younger collective (mean age: 64.3 years), which might have skewed the outcome parameters.

Until now, few studies have evaluated quality of life after intramedullary nailing of PHF. HRQoL with the EQ-5D-5L index/EurQol VAS patients of this study averaged 0.788 and 68.78, respectively. Compared to recent data published by Lopiz et al., these results can be viewed as very satisfactory. One must acknowledge that these data were generated out of an older population with a mean age of 82.1 years, recording an EQ-5D-5L index of 0.40 and EuroQolVAS 64.2 [3]. Yet, HRQoL values after IMN are lower than in a reference German population over 65 years (.84 ± 0.22 and 73.2 ± 18.5) [26].

Maurer et al. showed high correlations between PROMs and CMS (Constant–Murley Score) as well as strong to moderate correlation between PROMs, CMS, and HRQoL, which is also depicted in our results. In this sense, shoulder integrity plays an important role in the quality of life of patients and is a valid tool in patients after PHFs, as proposed by Olerud et al. [27]. This assertion is substantiated by the outcomes obtained from our findings.

### 4.2. Complications and Revisions

To date, only a limited number of studies have delved adequately into exploring the impact of complications and revisions on the clinical outcomes following antegrade nailing for PHFs. Notably, the influence of major complications and revisions on most of the PROMs appears to be minimal. This intriguing finding has the potential to empower surgeons when making decisions regarding revision surgery for surgically manageable complications, given that a substantial proportion of patients with major complications underwent successful revisions.

However, it is essential to highlight that the ASES assessment demonstrated the ability to identify a clinical decline in patients who encountered major postoperative complications and required revisions. These patients exhibited significantly poorer scores in comparison to those with either no complications or minor ones.

One plausible explanation for our findings may be linked to the ASES assessment’s particular focus on self-reported pain, which could strongly associate with postoperative complications and revision rates. Unlike the DASH questionnaire, where pain-related inquiries comprise only 6.67% (two questions), and the OSS, where they account for 25.00% (three questions), the ASES assessment incorporates a Pain Visual Analog Scale (VAS) that contributes to 50% of its overall scoring.

In support of our observations, Südkamp et al. reported similar results in their study involving 463 patients, wherein the CMS revealed significantly lower scores among patients with intraoperative complications, and non-significant lower scores in those with postoperative complications [28]. In contrast to our study, complications were not subclassified, which could have reduced the effect size in their data.

Zhu et al. assessed the functional outcome after surgical management of proximal fractures (using plating and nailing) in 57 patients using ASES. In contrast to our data, no significant influence of complications on ASES was established [13].

### 4.3. Fracture Complexity

It has previously been well published that four-part fractures pose a risk of complications and revisions [15,29]. These findings confirm our recent data with more than twice as many complications in four-part fractures compared to two- and three-part fractures. Furthermore, significantly more revisions were found.

The dataset in this study showed non-significant trends, suggesting that higher fragment counts were associated with inferior scores in PROMs and HRQol. Greenberg et al. conducted a study on the same interlocking nail (Targon PH+) and found a significant reduction in ASES scores for Neer 4 fractures compared to Neer 3 fractures [4].

Tsitsilonis et al. supported these findings in their work, when they evaluated a restricted sample of 43 patients using the CMS. Their investigation revealed significantly unfavorable outcomes for individuals with four-part fractures [30].

Interestingly, Kloub et al. conducted an analysis on a cohort of 137 patients diagnosed with three- and four-part fractures. Their findings indicated a marginal and non-significant disparity between the two groups in terms of CMSrel (Constant–Murley Score relative values) [31].

### 4.4. Age

Based on our data, age appears to be a critical factor influencing the clinical outcome following PHF. To assess the impact of age on PROMs and HRQol, we stratified our cohort into two subgroups: patients aged >65 and those aged <65 years. Remarkably, we identified significant differences in the measured PROMs and HRQol between these two age groups.

It has been noted that these results are very likely to be influenced by the high incidence of four-part fractures in patient over 65 years. On the contrary, a notable disparity in the incidence of complications and subsequent revisions became evident when comparing different age groups. Specifically, the younger age group exhibited a higher frequency of complications. It is worth noting that minor complications were particularly pronounced in patients under 65 years of age, which could be attributed to their potentially greater expectations and consequently, a higher tendency to report even minor incidents.

Our findings diverge from those reported by Dey Hazra et al., who reported no significant difference in clinical outcomes measured by PROMs when using the same age cutoff. Notably, Dey Hazra et al. focused predominantly on patients treated with plate osteosynthesis. Additionally, their inclusion of a substantial number of four-part fractures in the age group below 65 years might have impacted their results differently compared to ours [32].

In another study by Hättich et al., the influence of age on PROMs after antegrade nailing was investigated in a small sample size of 42. However, their data did not reveal a significant influence of age when using logistic regression analysis [29].

### 4.5. Limitations

Undoubtedly, this study possesses inherent limitations that warrant acknowledgment. Notably, a selection bias should be recognized due to a relatively high loss to follow-up. Patients at both extremes of the clinical spectrum demonstrate higher adherence to participation in clinical studies, with individuals who are either highly satisfied or dissatisfied showing a greater inclination to respond [33].

Moreover, the rigorous exclusion criteria, lengthy follow-up period, and the advanced mean age of the eligible pool of 260 patients resulted in the enrollment of only 70 individuals. Thus, future investigations should strive to encompass larger sample sizes.

Furthermore, it is important to note that this study was conducted solely at a single center characterized by a high level of expertise in proximal humeral nailing. To ascertain the generalizability of our findings, it is imperative to obtain data from multi-center investigations.

Nevertheless, despite these limitations, we firmly believe that this study provides valuable insights for clinicians and their patients.

## 5. Conclusions

This study provides valuable insights into proximal humeral fractures, offering a comprehensive understanding of multiple aspects, including risk factors, clinical outcomes, and treatment considerations related to nailing.

The clinical outcome measured by PROMs after interlocking nailing of proximal humeral fractures is considered good, consistent with the current literature. Notably, factors such as revision surgery, fracture complexity, and age appeared as influential elements, negatively impacting patient-reported outcome measure scores and well-being.

The study emphasized the significant correlation between PROMs and overall well-being, underscoring the critical role of shoulder integrity in health-related quality of life.

The current body of research on the impact of complications and revisions on clinical outcomes following antegrade nailing for PHFs remains limited. Nevertheless, our findings suggest that major complications and revisions seem to have minimal influence on most of the PROMs, potentially empowering surgeons to consider revision surgery for surgically manageable complications. However, it is crucial to acknowledge that the ASES assessment, with its focus on self-reported pain, was able to detect a decline in patients experiencing major postoperative complications and revisions, as evidenced by significantly lower scores compared to those with no or minor complications. This underscores the importance of comprehensive PROMs, which may help improve our understanding of patients’ experiences and guide surgical decision making in PHF management.

Fracture complexity appears to play a substantial role in the risk of complications and revisions, as indicated by the previous literature. Our recent data reinforces these findings, revealing a significant increase in complications in four-part fractures compared to two- and three-part fractures, along with a higher rate of revisions. Moreover, our dataset suggests non-significant trends, hinting that higher fragment counts might be associated with inferior scores in PROMs and HRQol. Although many studies have indicated unfavorable outcomes for patients with four-part fractures, the literature presents contrasting findings.

Despite this, it is important to consider the consistent evidence pointing to poorer outcomes associated with four-part fractures. As a result, there is a compelling need to explore alternative treatment approaches to address the challenges of four-part fractures effectively.

Although our data indicated an influence of age on PROMs and HRQoL, further investigation is required to fully understand the role of age in PHF considering the potential impact of confounding factors.

Despite limitations such as selection bias and limited sample size, this study provides valuable insights for clinicians and patients, calling for larger, multi-center investigations to enhance generalizability and deepen our understanding of proximal humeral fractures.

## Figures and Tables

**Figure 1 jpm-13-01224-f001:**
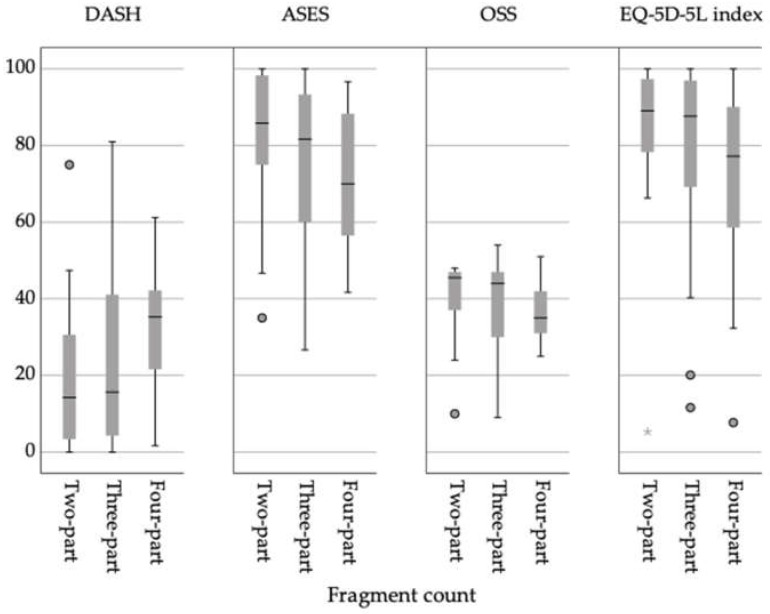
PROMs and HRQoL scores according to fragment count: *y*-axis displays absolute score values, and *x*-axis display sub-groups.

**Table 1 jpm-13-01224-t001:** Summary of Descriptive Statistics for Metric and Nominal Data. Metric data are reported with mean ± SD and range, nominal data with total number *n* and associated percentage (%).

	Mean ± SD	Range
Age	74.5 ± 11.6	(29–97)
Number of patients > 65 yrs.	58	-
Number of patients < 65 yrs.	12	-
Follow-up time	4.9 ± 1.9	(1.6–7.8)
	*n*	%
Sex		
Female	47	67.1%
Male	23	32.9%
Injured side		
Right	34	48.6%
Left	36	51.4%
Fracture parts		
Two-part	18	25.7%
Three-part	38	54.3%
Four-part	14	20.0%
Anatomical reduction	45	48.5%
Head–shaft-angle postoperatively		
	Mean ± SD	Range
Absolute	136.0° ± 9.9	(109.7–157.6)
	*n*	%
Varus < 130°	13	19.1%
Physiologic	33	48.5%
Valgus > 140°	22	32.4%

**Table 2 jpm-13-01224-t002:** Classification of Complication rate (CR)per Patient Case and Comparison between >65 and <65 years.

Complications	*n*	%	CR in >65 vs. <65 Years in% ***
Total	15	21.4%	15.4% vs. 41.7%
Major *	11	15.7%	13.8% vs. 25.0%
Minor **	4	5.7%	1.7% vs. 20.0%
Revision surgery	11	15.7%	13.8% vs. 25.0%

* Clavien–Dindo Grades III and IV; ** Calvien-Dindo Grades I and II; *** complication rates in the subgroups >65 and <65 years.

**Table 3 jpm-13-01224-t003:** Number of Complications and Associated Revisions. Some patients experienced ≥1 complications.

Complications	*n*	Related Revisions
Nail loosening	1	1
Cut-out	1	1
Loss of reduction	3	2
Humeral head necrosis	3	3
Prolonged pain	3	3
Joint stiffness	3	1
Prolonged consolidation	1	0
Keloid scar formation	2	0

**Table 4 jpm-13-01224-t004:** Overall means of PROMs and HRQoL; SD: standard deviation; *n* = 70.

	Mean ± SD	Range
DASH Score	25.4 ± 22.1	81.0–25.4
ASES Score	76.2 ± 21.1	100.0–76.2
OSS	38.8 ± 10.3	54.0–9.0
EQ-5D-5L index value	0.79 ± 0.23	1.00–0.05
EuroQol VAS	68.8 ± 17.9	35.0–100.0

**Table 5 jpm-13-01224-t005:** Correlation Matrix of DASH, ASES, OSS, and EQ-5D-5L Scores.

		DASH	ASES	OSS	EQ-5D-5L
DASH Score	Pearson correlation	-	−0.812 *	−0.902 *	−0.823 *
	Sig. (2-tailed)		0.000	0.000	0.000
ASES Score	Pearson correlation	−0.813 *	-	0.852 *	0.541 *
	Sig. (2-tailed)	0.000		0.000	0.000
OSS	Pearson correlation	−0.902 *	0.852 *	-	0.722 *
	Sig. (2-tailed)	0.000	0.000		0.000
EQ-5D-5L	Pearson correlation	−0.823 *	0.541 *	0.722 *	-
	Sig. (2-tailed)	0.000	0.000	0.000	

* The correlation is highly significant at the level of 0.01 (2-tailed).

**Table 6 jpm-13-01224-t006:** PROMs and HRQoL subdivided into fragment count. Displayed via mean ± SD (range).

Outcome Measure	Two-Part *n* = 18	Three-Part *n* = 38	Four-Part *n* = 14
DASH	20.9 ± 21.4 (0.0–75.0)	25.1 ± 23.7 (0.0–81.0)	32.3 ± 17.1 (1.7– 61.2)
ASES	81.1 ± 19.9 (35.01–100.00)	76.4 ± 22.5 (26.66–100)	69.5 ± 18.0 (41.65–55.01)
OSS	41.0 ± 10.0 (10.0–48.0)	39.0 ± 11.0 (9.0–54.0)	36.0 ± 8.0 (25.0–51.0)
EQ-5D-5L index	0.830 ± 0.221 (0.053–1.00)	0.802 ± 0.222 (0.116–1.00)	0.699 ± 0.278 (0.077–1.00)
EuroQol VAS	66.0 ± 18.0 (40.0–100.0)	70.0 ± 19.0 (35.0–100.0)	70.0 ± 17.0 (50.0–100.0)

## Data Availability

Not applicable.

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
