# Peer review of "Risk Factors and Clinical Outcomes after Antegrade Intramedullary Nailing in Proximal Humeral Fractures: Insights and Implications for Patient Satisfaction"

_jpm, 2023, doi:10.3390/jpm13081224_

Round 1
Reviewer 1 Report
In the submitted manuscript, the authors analyzed risk factors and clinical outcomes after antegrade intramedullary nailing (using the Targon PH+ intramedullary nail by Fa. Aesculap, Germany) in proximal humeral fractures (PHF). The retrospective study provides insights and implications for PHF patient satisfaction. The main aims and the need for this study are clearly mentioned in the introduction. The study design is well elaborated, where the authors included a subset of 70 patients in the study and collected and analyzed data, including radiologic fracture assessment (plain radiographic images or CT scans), evaluation of complications and PROMs (DASH, OSS, ASES), and HRQoL (EQ-5D-5L, EuroQol VAS) study questionnaires. The study concluded that the use of the Targon PH+ intramedullary nail for the treatment of PHFs resulted in satisfactory clinical outcomes and acceptable complication and revision rates. The authors also mention that the outcomes of this study align harmoniously with the existing scientific literature.
Overall, the manuscript is well-written. The conclusions are substantiated by the data, and the discussion touches on some limitations of the study, related work in the field, and highlights the importance of the findings. However, there are a few areas where minor changes might be necessary.
1. On line 213: Results of PROMs and HRQoL measures should be referred to the appropriate table (I suppose Table 4) added to the manuscript.
2. On line 219: Correlation results should be referred to the appropriate table (I suppose Table 5) added to the manuscript.
3. On line 225: The mentioned Figure 2 is missing in the manuscript. I suppose the authors mean Figure 1 here. Kindly refer to the appropriate figure.
4. On line 225: The bracket needs to be closed.
5. Figure 1: The values presented on the Y-axis may be in percentage. If so, authors should specify that.
Author Response
Dear Reviewer,
We hope this finds you well. We are writing to express our sincere gratitude for your time and effort in thoroughly reviewing our manuscript titled and providing us with insightful feedback.
We are pleased to inform you that we have carefully considered all your comments and suggestions, and have made the necessary revisions accordingly.
Regarding your specific points of concern:
-
On line 213: We have now referred to the results of PROMs and HRQoL measures in the appropriate table, which is Table 4.
-
On line 219: The correlation results have been correctly referenced in Table 5 as you suggested.
-
On line 225: We have rectified the error and referred to the correct figure (Figure 1) in the text.
-
On line 225: We have ensured that the bracket has been closed appropriately.
-
Figure 1: As you pointed out, we have specified in the figure caption that the values presented on the Y-axis are the score's absolute values.
Once again, thank you for your time and support in reviewing our manuscript. We are truly grateful for your commitment to maintaining the quality of scientific literature.
Please let us know if you require any further information or have any additional suggestions.
With warm regards,
Maximilian Willauschus
Reviewer 2 Report
In this article, the authors presented statistical data on the potential risks and clinical outcomes of antegrade intramedullary nailing in patients with proximal humeral fractures. A large enough data has been considered while assessing patient satisfaction. Overall understanding of the paper is reasonably good. However, the paper lacks scientific comprehension. This manuscript is acceptable following "major revision". To improve the quality of the manuscript the review comments are appended below.
1) The opening section should first and foremost make it apparent what the main originality of this study is. What are the most recent humeral fracture statistics? What age groups are most likely to suffer a humeral fracture? Then, it's important to give the readers some fundamental knowledge about the surgical intervention process in order to encourage further reading. Additionally, it's crucial to identify the humeral fracture's clinical outcome and to give a brief overview of the length of the course of treatment.
2) The basic concepts of the antegrade intramedullary nailing technique should also be covered in the introduction part. When did this approach begin? Exactly what kind of nails are being used? It is also beneficial to describe the nail's biomechanics in both stationary and moving situations. What might be the dangers in both standing and moving situations. What circumstances necessitate revision surgery? Additionally, it is beneficial to advance scientific knowledge regarding how the weight will be distributed in cases of two, three, or four fractures. In case of four fractures which zone is more prone to failure after antegrade intramedullary nailing. Why not try to reason it out? If you have sufficient case history data, please include it.
3) The nail material which is the most crucial piece of information is absent. The oxidative stress mediators would be rather high right after the fracture, which could result in implant material leaching due to low pH (approximately 3 to 3.5). For instance, the release of Ni from an SS316L implant could result in severe pathophysiological issues by activating proinflammatory cytokines. Moreover, SS316L is not tissue friendly compared to Ti. Additionally, the weight density of SS316L would be somewhat higher than the human bone, which would result in stress shielding and delayed wound healing, while in the case of Ti-6Al-4V alloy, the dissolution of Al in blood stream could result in an allergic reaction and an aggravation of Alzheimer's disease. Therefore, one of the key determining elements for the clinical outcome of the patient is the implant material. I find it surprising that information has been ruled out in this study. It's a good idea to mention the nail's composition. Regardless of whether it is made of titanium alloy (Ti-6Al-4V), pure titanium, or stainless steel 316L. I encourage you reading the following papers to better understand how implant material affects osteointegration.
https://doi.org/10.1016/j.jallcom.2023.169852
https://doi.org/10.1016/j.matpr.2022.05.469
4) A large age range (29 to 97) was taken into account in Table.1. How many of these groups are elderly? As is well known, a significant portion of the aged population suffers from joint failure. To understand the age-specific clinical result, it is necessary to precisely discriminate between the young and old groups.
5) The Table.2 presents the information of major and minor complications. It is good to point out what are the age groups they are prominent to major and minor complications. Also, it is good to reason out why such complications arises and what are the causative factors for it? It is also good to provide information that within how many days of surgery minor and major complications most likely to onset?
6) Table.3 lists elements like nail loosening. However, it was not mentioned whether the nail loosening was brought on by a stress shielding effect, an implant-related infection, or an increase in pro-inflammatory cytokines in the implanted zone. The causes of cut-out, loss of reduction, humeral head necrosis, prolonged discomfort, joint stiffness, prolonged consolidation, and keloid scar formation should be discussed for the readers' better understanding. Verifying whether any cases among these are diabetic.
7) Conclusions, which should include a thorough overview of each part, can be much more illuminating.
9) If possible please include interactive images of the implant, the zone of implant fixation, and bone failures to reach a wider audience.
10) There are a few typos and grammatical mistakes that need to be double checked.
11) Table and figure captions should be revised in a simple understandable way.
Author Response
Dear Reviewer,
We sincerely appreciate your thoughtful review of our manuscript. Your comments and suggestions have provided valuable insights to enhance the scientific comprehension and overall quality of the paper. We have diligently addressed each of your concerns and made significant revisions accordingly. Below, we present a point-by-point response to the issues you raised:
Introduction:
We have thoroughly revised the opening section to clearly state the main originality of this study. We now provide the historical development, the types of nails utilized, and a discussion on the biomechanics and also highlighted the age groups most susceptible to humeral fractures.
Antegrade Intramedullary Nailing Technique:
In response to your suggestion, we refer to the method section where the reader finds detailed explanation of the antegrade intramedullary nailing technique used. We also added some information to the postoperative care.
Nail Material and Implant Composition:
While we agree with the significance of discussing the impact of implant materials, we encountered a limitation in our study concerning this aspect. Unfortunately, due to the lack of comprehensive data on the local release of nail composites and their role in local inflammation, a thorough discussion on the specific implications of material choice was not possible within the scope of this study. Consequently, the potential influence of implant material on osteointegration and clinical outcomes remains speculative in our study.
We believe that investigating the impact of various implant materials on patient outcomes is indeed an essential area of research. However, it requires extensive and dedicated studies with a larger sample size and specific failure analysis to draw definitive conclusions. We appreciate your suggestion and agree that understanding the influence of implant material on the success of antegrade intramedullary nailing is a topic that warrants further investigation in future research endeavors.
Age Range:
We have carefully analyzed our cohort and divided the age groups into young (<65 years of age) and elderly (>64 years of age) categories, enabling a more accurate assessment of age-specific clinical outcomes. We included the new data into the results section and into the new discussion.
Complications:
In Table 2, we have provided a detailed breakdown of the data concerning the specified age groups, addressing the concerns raised earlier. Additionally, we have included a comprehensive discussion on the potential causes of various complications observed during the study.
Table 3: We sincerely appreciate your valuable suggestion to delve deeper into the causes of nail loosening and other complications. Regrettably, our study's scope did not encompass a thorough analysis of each specific cause of complication or implant failure, and as a result, we lack specific data to provide a detailed exploration. Nonetheless, we acknowledge the importance of this aspect and recommend it as a potential avenue for future research in this field.
Conclusions:
The Conclusions section has been thoroughly revised to offer a more comprehensive overview of each part, providing readers with clearer insights into our findings.
Interactive Images:
Unfortunately, due to the limitations of the current publishing format, we are unable to include interactive images.
Typos and Grammar:
We have conducted a meticulous proofreading process and have rectified all identified typos and grammatical errors in the revised manuscript.
Table and Figure Captions:
We have rephrased the captions for all tables and figures to ensure they are simple, concise, and easily understandable.
Once again, we express our sincere gratitude for your valuable feedback and the opportunity to improve our manuscript. We are confident that the revised version now meets the required scientific standards and comprehensibility. We look forward to your evaluation of the updated manuscript.
Thank you for your time and consideration.
In the name of all authors
Maximilian Willauschus